

# A cosine adaptive particle swarm optimization based long-short term memory method for urban green area prediction

Hao Tian[1,2], Hao Yuan[2], Ke Yan[3] and Jia Guo[1,2,4]

[1] Hubei Key Laboratory of Digital Finance Innovation (Hubei University of Economics), Wuhan, China
[2] School of Information Engineering, Hubei University of Economics, Wuhan, Hubei, China
[3] China Construction Third Engineering Bureau Installation Engineering Co., Ltd., Wuhan, Hubei, China
[4] Hubei Internet Finance Information Engineering Technology Research Center, Hubei University of Economics, Wuhan, China

Corresponding author
Jia Guo, guojia314@gmail.com

## ABSTRACT

In the quest for sustainable urban development, precise quantification of urban green space is paramount. This research delineates the implementation of a Cosine Adaptive Particle Swarm Optimization Long Short-Term Memory (CAPSO-LSTM) model, utilizing a comprehensive dataset from Beijing (1998–2021) to train and test the model. The CAPSO-LSTM model, which integrates a cosine adaptive mechanism into particle swarm optimization, advances the optimization of long short-term memory (LSTM) network hyperparameters. Comparative analyses are conducted against conventional LSTM and Partical Swarm Optimization (PSO)-LSTM frameworks, employing mean absolute error (MAE), root mean square error (RMSE), and mean absolute percentage error (MAPE) as evaluative benchmarks. The findings indicate that the CAPSO-LSTM model exhibits a substantial improvement in prediction accuracy over the LSTM model, manifesting as a 66.33% decrease in MAE, a 73.78% decrease in RMSE, and a 57.14% decrease in MAPE. Similarly, when compared to the PSO-LSTM model, the CAPSO-LSTM model demonstrates a 58.36% decrease in MAE, a 65.39% decrease in RMSE, and a 50% decrease in MAPE. These results underscore the efficacy of the CAPSO-LSTM model in enhancing urban green space area prediction, suggesting its significant potential for aiding urban planning and environmental policy formulation.

# INTRODUCTION

Urban green spaces manifest as critical elements within the urban fabric, contributing significantly to both ecological sustainability and human well-being. These verdant areas offer a multitude of benefits, ranging from environmental amelioration to social and psychological advantages. Ecologically, urban greenery plays a pivotal role in enhancing biodiversity, providing habitats for various species, and maintaining ecological balance

within urban ecosystems. This biodiversity is not only vital for ecosystem health but also contributes to the resilience of urban areas against environmental stressors. *Rao et al. (2022)* evaluates the accessibility of urban green spaces in 254 Chinese cities using the two-step floating catchment area method, revealing significant disparities in accessibility both within and between cities, especially within walking or cycling distances. It associates higher social status, reflected in housing prices, with greater access to green space services, highlighting pronounced inequities. The research offers targeted recommendations to improve the distribution and use of urban green spaces, aiming to enhance their ecological, social, and economic benefits in urban ecosystems.

From an environmental perspective, green spaces are instrumental in mitigating urban pollution. *Ghahramani et al. (2021)* undertook a novel application of Artificial Intelligence (AI) techniques in the realm of urban green spaces (UGS) assessment by implementing a unified topic modeling approach. Investment in park green space can improve the quality of life for urban residents. *Paul et al. (2020)* conducted a study on large cohorts in Ontario, Canada that found that increased exposure to urban green space is associated with a reduced risk of developing major neurological conditions, specifically dementia and stroke. *Stuhlmacher, Kim & Kim (2022)* conducted a study to analyze the relationship between the development of park and non-park green spaces and the likelihood of gentrification in Chicago, using satellite imagery and demographic data, revealing that green space investments' impact on gentrification varies with time and neighborhood characteristics. *Hu et al. (2022)* investigates the relationship between residential green space, neighborhood walkability, and atherosclerosis in urban settings, analyzing data from 2021 adults with suspected coronary heart disease (CHD). Utilizing advanced statistical methods, the study evaluates the impact of green space and walkability on coronary artery calcium scores, highlighting the complex interplay between urban development, environmental factors, and cardiovascular health. *Hogendorf et al. (2020)* found that while increasing distance to green spaces slightly reduced leisure walking time and slightly increased walking for active commutes among Dutch adults, there was no significant association with cycling, indicating a limited impact of green space proximity on walking and cycling behaviors. *Basu & Nagendra (2021)* found that public spaces like parks exhibit significant gender and income inequalities, resulting in uneven access to green space. *Jiang, Stickley & Ueda (2021)* explored the relationship between green spaces and suicide mortality in Japan, revealing that the protective impact of greenery against suicide varies with urbanity and demographic factors, indicating the potential of green spaces in suicide prevention strategies.

## RELATED WORK

The development of urban green spaces is influenced by various factors. *Kwartnik-Pruc & Trembecka (2021)* examines the involvement of local governments in shaping public green spaces, essential to sustainable urban development, with a specific focus on Poland. It analyzes municipal strategies in property acquisition, legal frameworks, and policy formulation for green space enhancement, using data from the Polish Central Statistical Office to track the progression of public green spaces in major Polish cities, notably

Krakow. The study not only (*Aly & Dimitrijevic, 2022*) investigated the distribution and types of green spaces in Krakow but also assessed the municipality's initiatives in expanding and safeguarding these areas against urban development, emphasizing the significance of green space accessibility for residents. The findings provide a comprehensive evaluation of Krakow's management of green spaces and reflect broader trends in public green space development, offering valuable insights for future urban environmental policies and sustainable planning practices. *Feltynowski & Kronenberg (2020)* investigates the proportion and types of urban green spaces in five towns in Poland, comparing various data sources to reveal discrepancies in the perception and management of green spaces, especially highlighting the underestimation of green areas in smaller towns when relying solely on public statistics.

Long short-term memory (LSTM) models are used in traffic flow prediction (*Bharti, Redhu & Kumar, 2023*). In 2020, *Stessens et al. (2020)* developed a GIS-based model to infer the perceptions of naturalness, quietness, and spaciousness by users of public green spaces based on the attributes of the green spaces. *Das (2022)* examined factors contributing to the environmentally unjust development and management of organized green spaces in three Indian cities (Bhubaneswar, Cuttack, and Kolkata), and evaluated various strategies aimed at achieving environmental justice. *Huang, Wu & Cheng (2021)* demonstrated that a strategically designed ecological network effectively augments landscape connectivity and curtails fragmentation, consequently elevating the quality of the urban ecological environment and fostering the sustainability of urban green spaces.

Predicting the area of urban green spaces is critically important for several interconnected reasons. Accurate green space predictions are vital for public health and well-being, as these areas provide essential spaces for recreation, relaxation, and socializing, thereby enhancing the mental and physical health of city residents. Long short-term memory models are often used for predicting tasks like landslide displacement (*Duan, Su & Fu, 2023*) and air pollutant prediction (*Luo & Gong, 2023*). *Zhou, Zuo & Zhao (2022)* proposed a large-scale urban land subsidence prediction method. *Usharani (2022)* introduced an improved loss function within an LSTM neural network to enhance the accuracy of sea surface temperature predictions at various time horizons around India. *Luo & Gong (2023)* developed an innovative ARIMA-WOA-LSTM model, aimed at enhancing the precision in predicting air pollutants, thereby optimizing air pollution management. By employing ARIMA to process the linear components of pollution data and utilizing a whale optimization algorithm-enhanced LSTM (WOA-LSTM) for predicting non-linear elements, the research significantly advances the model's efficacy. The study further substantiates the superiority of the ARIMA-WOA-LSTM model over various counterparts in terms of pollutant prediction accuracy, model prediction precision, and stability, through comparative analyses with several other models. *Du et al. (2022)* proposed a Particle Swarm Optimization (PSO) based LSTM model for urban water demand forecasting.

This has also catalyzed research on enhanced particle swarm algorithms. Metaheuristics methods (*Aranha et al., 2022*; *Sörensen, 2015*) based on a diverse range of natural, artificial, and social behaviors or patterns. These methods are often used to solve optimization

problems in various fields (*Velasco, Guerrero & Hospitaler, 2022*, *2024*). For instance, *Guo et al. (2024)* proposed a novel dynamic learning method for breast cancer image classification. In 2022, *Guo et al. (2022)*, *Tian et al. (2022)* discussed novel strategies for swarm intelligence. In 2023 and 2024, novel metaheuristic methods (*Guo et al., 2023a*, *2023b*; *Zhou et al., 2024*) are proposed for single-objective optimization problems.

To increase the accuracy of prediction, this article presents a novel cosine adaptive particle swarm optimization-based long short-term memory model for urban green spaces prediction. The major contribution of this article can be summarized as follows:

1. A new cosine adaptive strategy applied to the evolution of particle swarm algorithms enhances the search capability of traditional particle swarm algorithms. The cosine adaptive strategy enhances the particle swarm's ability to escape local optima and increases the diversity of the particle swarm.

2. The Cosine Adaptive Particle Swarm Optimization (CAPSO)-LSTM is used in urban green area prediction, and experimental results demonstrate that the proposed CAPSO-LSTM can accurately predict the area of urban green spaces, providing significant assistance in urban construction planning.

## MATERIALS AND METHODS

### Cosine adaptive particle swarm optimization

The particle swarm optimization (PSO) algorithm has the characteristics of strong generality and simple principle and has received a lot of attention from researchers since it was proposed. However, the algorithm also has some drawbacks, such as easy to fall into local optimal. To address these shortcomings, we propose the CAPSO, in which the velocity update formula of particles in the algorithm is consistent with the PSO algorithm. But when updating the positions of the particles, we add an adaptive mechanism to it, which can achieve a better balance between exploration and exploitation. In the CAPSO, the velocity and position of the particle will be updated by Eq. (1).

$$
\begin{aligned}
v_{t+1}(i) &= w * v_t(i) + c_1 * r * (pbest_t(i) - x_t(i)) + c_2 * r * (gbest_t - x_t(i)) \\
x_{t+1}(i) &= x_t(i) + (1 + \cos(i * \Pi/(\exp(i)))) * v_t(i)
\end{aligned}
\tag{1}
$$

where $v_{t+1}(i)$ is the velocity of the *ith* particle at the $t + 1$ iteration, $w$ is the weight of the velocity, $c_1$ and $c_2$ are the learning factor of the particle, r is a random number from 0 to 1, which can improve the randomness of the search. In the t iteration, *pbest* and *gbest* represent the position of the particle and the best position of the global particle respectively. exp() is the oscillation coefficient of the t evolution and cos() is the cosine function.

### Long short-term memory cosine adaptive particle swarm optimization

Long short-term memory (LSTM) networks, a pivotal subset of recurrent neural networks (RNNs), are designed to overcome the limitations of traditional RNNs in learning long-term dependencies. Distinct for their unique architecture, LSTMs are equipped with specialized units called gates: the forget gate, input gate, and output gate. These gates
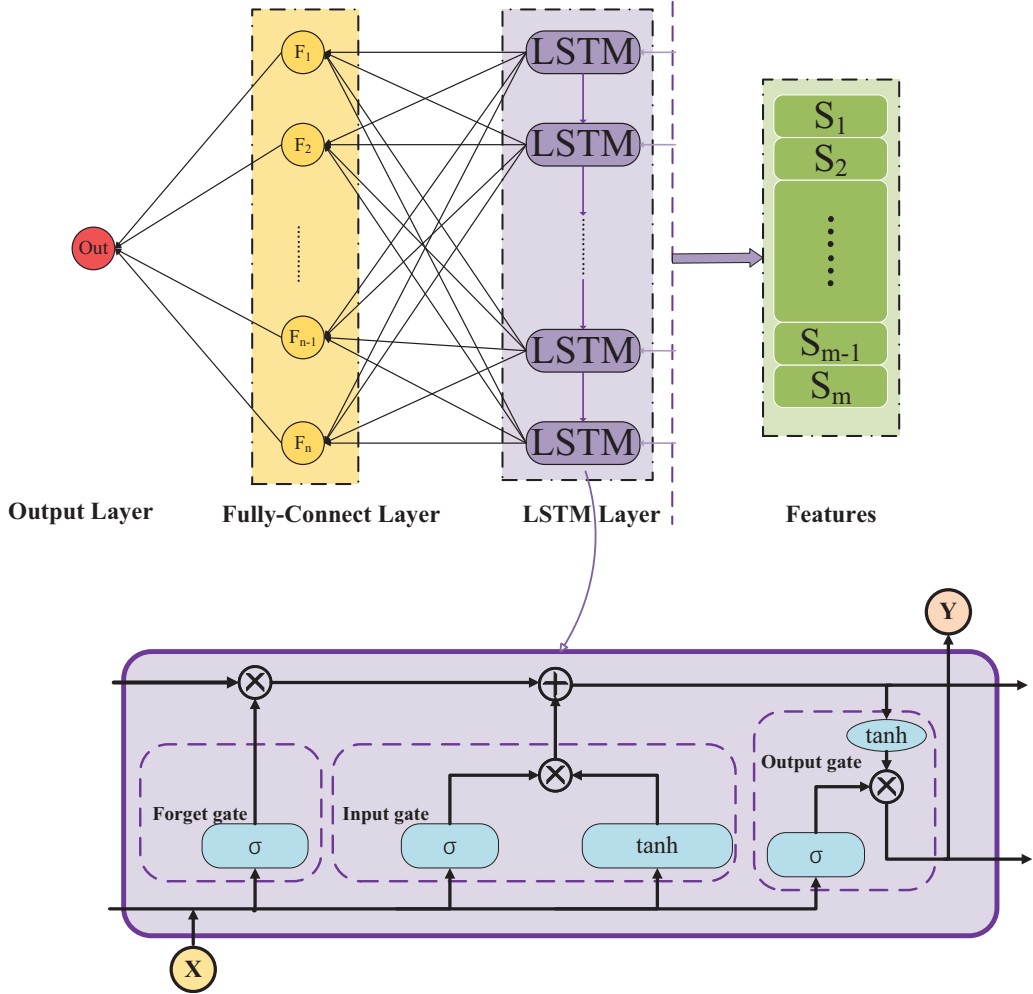

**Figure 1** **Architecture diagram LSTM.** The image depicts an LSTM (long short-term memory) neural network architecture, illustrating the flow and transformation of data within. It shows the internal gating mechanisms—forget, input, and output gates—of an LSTM cell, how they process the input X, and generate an output Y. The LSTM layer connects to a fully connected layer that integrates the features, leading to the final output layer where the result is produced.

effectively regulate the flow of information, allowing the network to retain or discard data over intervals of varying lengths. This capability is particularly beneficial in tasks involving sequential data, such as natural language processing, time series analysis, and speech recognition. By addressing the issue of vanishing gradients, a common problem in standard RNNs, LSTMs facilitate more efficient and robust learning of dependencies across extensive time lags, making them a fundamental tool in the field of deep learning. The architecture diagram of LSTM is shown in Fig. 1.

In the domain of LSTM neural networks, it has been demonstrated that the accuracy of prediction outcomes can be enhanced through the modification of the learning rate and the neuronal count. The CAPSO-LSTM algorithm is introduced, representing an amalgamation of LSTM with the CAPSO algorithm. This integration is achieved by

employing CAPSO to optimize both the learning rate and the neuron quantity within the LSTM framework, thereby augmenting the precision of the prediction model. The algorithm's methodology encompasses the following specific steps:

- Step 1: Normalization is performed by Eq. (2), which ranges from 1 to −1. Normalization eliminates the undesirable effects caused by odd sample data and improves the convergence of the training network.

$$X' = 2 * (X - \min(x))/(\max(x) - \min(x)) - 1 \qquad (2)$$

where X is the input data, min(x) is the minimum value of input data, max(x) is the maximum value of input data, X′ is the result after normalization.

- Step 2: In the CAPSO algorithm part, the optimization objects are the initial **learning rate** and the **number of neurons** of LSTM, and the fitness function is a mean square error (MSE). The fitness function of the CAPSO is shown in Eq. (3):

$$MSE = \frac{1}{n} \sum_{i=1}^{n} (Y_i - Y_i')^2 \qquad (3)$$

where $n$ is the number of samples, The $Y_i$ is the true value and $Y_i'$ is the predicted value.

- Step 3: In the LSTM training process, the optimal number of neurons and the initial learning rate obtained by the CAPSO algorithm will be used as parameters in the LSTM.
- Step 4: Start model prediction and calculate model prediction error.

The flowchart of CAPSO-LSTM is shown on Fig. 2.

# EXPERIMENTS

## Data preparation

Urban green spaces are correlated with several socio-economic and environmental factors. These include the local population size, the area allocated to park green spaces, the overall water supply provision, the expanse of road infrastructure, and the urban Gross Domestic Product (GDP). For the purpose of forecasting the area covered by urban green spaces, a dataset spanning 24 years (1998 to 2021) pertaining to Beijing has been extracted from the China Statistical Yearbook. Seven distinct indicators have been identified as inputs for the predictive model: General public budget revenue (PBR), general public budget expenditure (PBE), household population (HP), park green area (PGA), total water supply (TWS), road area (RA), and GDP, along with the measured green area (GA). These datasets are delineated in Tables 1 and 2. The temporal division of the dataset assigns the years 1998 to 2016 for the training phase of the model, whereas data from 2017 to 2021 are reserved for the validation of the predictive accuracy of the model.

To validate the predictive superiority of the CAPSO-LSTM model for urban green space quantification, the LSTM and PSO-LSTM models are utilized as baselines within the control group of the experiment. The uniformity in the selection of parameters and the delimitation of the optimization objectives for each model was preserved to ensure the

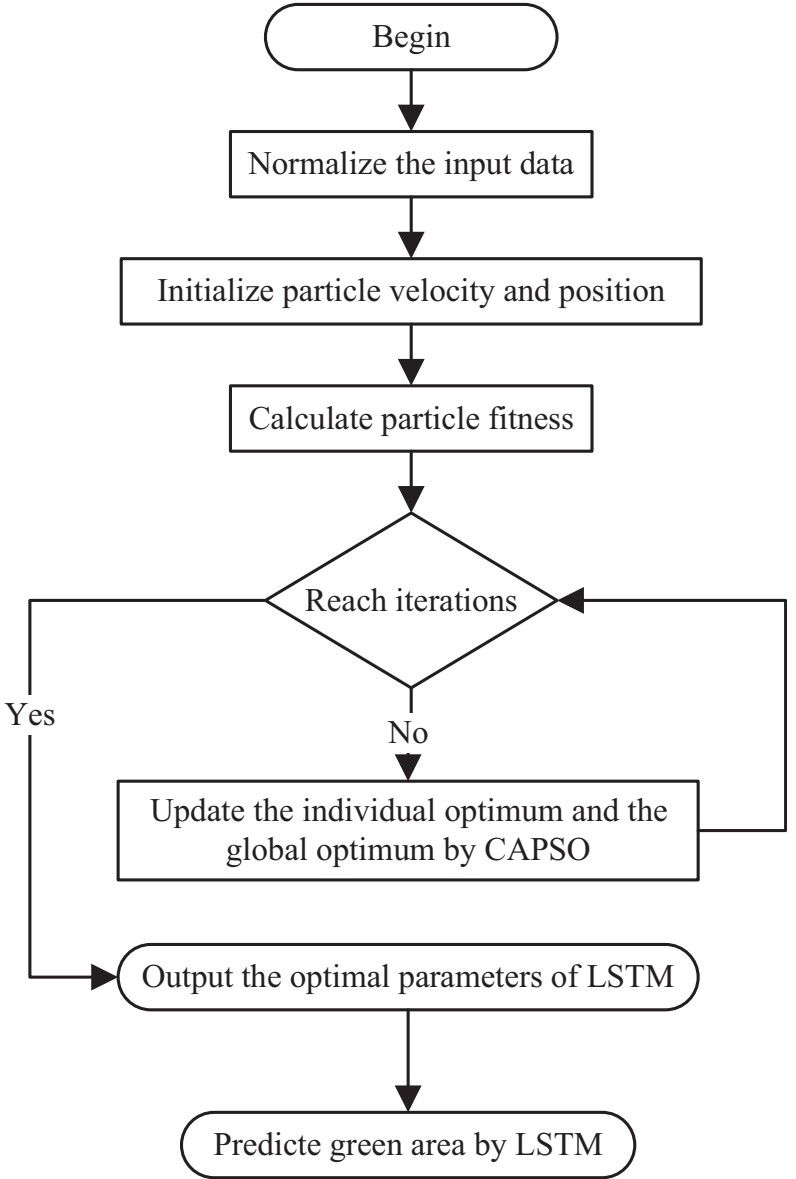

**Figure 2 Flowchart of CAPSO-LSTM.** The process begins with normalizing the input data, followed by initializing the particle velocity and position. It then calculates the particle fitness and checks if the predefined number of iterations has been reached. If not, it updates the individual and global optima using CAPSO. This loop continues until the iteration condition is met. Once completed, the process outputs the optimal parameters for the LSTM model, which are then used to predict the green area. The flow is sequential and iterative, with a decision point that loops back until the stopping criterion is satisfied.                   

integrity and comparability of the experimental outcomes, as detailed in Table 3 and follows.

   Number of particles (50): The number of particles determines the coverage of the search space. Choosing 50 particles represents a compromise between computational resources and search efficiency. Fewer particles might not explore the search space sufficiently, while

**Table 1 Specific data of public budget revenue (PBR), general public budget expenditure (PBE), household population (HP), park green area (PGA).**

| Indicators | PBR | PBE | HP | PGA |
|---|---|---|---|---|
| Unit | Billion yuan | Billion yuan | Peaple | Square hectometer |
| 1998 | 1.3345E+06 | 1.3460E+06 | 9.8186E+06 | 4.9540E+03 |
| 1999 | 1.0692E+06 | 1.4747E+06 | 1.0527E+07 | 4.9890E+03 |
| 2000 | 1.5755E+06 | 1.8829E+06 | 9.7414E+06 | 5.5130E+03 |
| 2001 | 4.5417E+06 | 5.5911E+06 | 9.8810E+06 | 7.0697E+03 |
| 2002 | 5.2433E+06 | 6.0728E+06 | 1.0670E+07 | 9.5771E+03 |
| 2003 | 5.8155E+06 | 7.0580E+06 | 1.0792E+07 | 1.0826E+04 |
| 2004 | 7.3159E+06 | 8.6225E+06 | 1.0929E+07 | 1.2446E+04 |
| 2005 | 9.0897E+06 | 1.0174E+07 | 1.1106E+07 | 1.1365E+04 |
| 2006 | 1.1061E+07 | 1.2476E+07 | 1.1269E+07 | 1.4234E+04 |
| 2007 | 1.4778E+07 | 1.5942E+07 | 1.1454E+07 | 1.1821E+04 |
| 2008 | 1.8216E+07 | 1.8941E+07 | 1.1611E+07 | 1.2316E+04 |
| 2009 | 2.0075E+07 | 2.2192E+07 | 1.1763E+07 | 1.8070E+04 |
| 2010 | 2.3312E+07 | 2.6119E+07 | 1.1910E+07 | 1.9020E+04 |
| 2011 | 2.9787E+07 | 3.1239E+07 | 1.2101E+07 | 1.9728E+04 |
| 2012 | 3.2838E+07 | 3.5487E+07 | 1.2291E+07 | 2.1178E+04 |
| 2013 | 3.6257E+07 | 4.0050E+07 | 1.2467E+07 | 2.3223E+04 |
| 2014 | 3.9883E+07 | 4.3550E+07 | 1.2631E+07 | 2.3223E+04 |
| 2015 | 4.7239E+07 | 5.7377E+07 | 1.2737E+07 | 2.9503E+04 |
| 2016 | 5.0813E+07 | 6.4067E+07 | 1.3595E+07 | 3.0069E+04 |
| 2017 | 5.4308E+07 | 6.8195E+07 | 1.3605E+07 | 3.1019E+04 |
| 2018 | 5.7859E+07 | 7.4675E+07 | 1.3737E+07 | 3.2619E+04 |
| 2019 | 5.8171E+07 | 7.4083E+07 | 1.3921E+07 | 3.5157E+04 |
| 2020 | 5.4839E+07 | 7.1162E+07 | 1.3955E+07 | 3.5720E+04 |
| 2021 | 5.9323E+07 | 7.2051E+07 | 1.4088E+07 | 3.6397E+04 |

more particles increase computational costs. Fifty particles are considered as a reasonable number to provide good search coverage within a reasonable timeframe.

Number of iterations (100): The number of iterations determines the length of the optimization process. One hundred iterations allow the PSO and CAPSO algorithms enough time to adjust the positions of its particles to find the optimal solution. This figure is based on experimental tuning and experience, aiming to balance between convergence speed and computational cost.

Range of neurons (10, 50): The number of neurons directly affects the model's complexity and capacity. Too few neurons might lead to underfitting, while too many neurons can cause overfitting and unnecessary computational burden. Choosing a range of 10 to 50 offers sufficient flexibility to find an optimal balance between performance and complexity.

Range of learning rates (0.001, 0.15): The learning rate is a key hyperparameter that determines the speed at which a model learns. A smaller learning rate (*e.g.*, 0.001) ensures

**Table 2 Specific data of total water supply (TWS), road area (RA), gross domestic product (GDP), and the green area (GA).**

| Indicators | TWS | RA | GDP | GA |
|---|---|---|---|---|
| Unit | Million cubic meters | Million cubic meters | Million yuan | Square hectometer |
| 1998 | 1.0905E+05 | 3.6206E+03 | 1.7810E+07 | 1.8682E+04 |
| 1999 | 1.1074E+05 | 3.6850E+03 | 1.8246E+07 | 1.9070E+04 |
| 2000 | 1.0934E+05 | 4.1988E+03 | 2.3323E+07 | 2.0600E+04 |
| 2001 | 1.4091E+05 | 5.9168E+03 | 2.6979E+07 | 2.9365E+04 |
| 2002 | 1.3899E+05 | 7.6450E+03 | 3.1245E+07 | 4.2592E+04 |
| 2003 | 1.2882E+05 | 1.0570E+04 | 3.5573E+07 | 4.8496E+04 |
| 2004 | 1.5021E+05 | 1.1213E+04 | 4.1610E+07 | 4.9298E+04 |
| 2005 | 1.4476E+05 | 1.6227E+04 | 6.7656E+07 | 4.4384E+04 |
| 2006 | 1.4264E+05 | 9.8580E+03 | 7.7374E+07 | 5.3163E+04 |
| 2007 | 1.4263E+05 | 7.7340E+03 | 9.2076E+07 | 4.4840E+04 |
| 2008 | 1.4251E+05 | 8.9410E+03 | 1.0325E+08 | 4.6993E+04 |
| 2009 | 1.5182E+05 | 9.1790E+03 | 1.1972E+08 | 6.1695E+04 |
| 2010 | 1.5556E+05 | 9.3950E+03 | 1.3904E+08 | 6.2672E+04 |
| 2011 | 1.5836E+05 | 9.1640E+03 | 1.6014E+08 | 6.3540E+04 |
| 2012 | 1.5965E+05 | 1.3509E+04 | 1.7617E+08 | 6.5540E+04 |
| 2013 | 1.8748E+05 | 1.3884E+04 | 1.9213E+08 | 6.8438E+04 |
| 2014 | 1.8242E+05 | 1.3834E+04 | 2.1019E+08 | 6.8438E+04 |
| 2015 | 1.8252E+05 | 1.4302E+04 | 2.3015E+08 | 8.1305E+04 |
| 2016 | 1.9137E+05 | 1.4316E+04 | 2.5669E+08 | 8.2113E+04 |
| 2017 | 1.8828E+05 | 1.3960E+04 | 2.8015E+08 | 8.3501E+04 |
| 2018 | 1.9198E+05 | 1.4098E+04 | 3.0320E+08 | 8.5286E+04 |
| 2019 | 1.5767E+05 | 1.4318E+04 | 3.5371E+08 | 8.8704E+04 |
| 2020 | 1.4779E+05 | 1.4702E+04 | 3.6103E+08 | 9.2683E+04 |
| 2021 | 1.5014E+05 | 1.4800E+04 | 4.0270E+08 | 9.3127E+04 |

**Table 3 Parameters in experiments of the particle swarm optimization (PSO) and cosine adaptive PSO (CAPSO).**

| Parameter | PSO | CAPSO |
|---|---|---|
| Populations | 50 | 50 |
| Iterations | 100 | 100 |
| Range of neurons | (10, 50) | (10, 50) |
| Range of learning rate | (0.001, 0.15) | (0.001, 0.15) |

stability in the learning process but may require more time to converge. A larger learning rate (*e.g.*, 0.15) can accelerate the learning process but might also lead to overshooting and instability. This range is chosen to provide enough flexibility for the PSO and CAPSO algorithms to find the best balance between stability and convergence speed.

**Table 4 Prediction results of LSTM, PSO-LSTM and CAPSO-LSTM.**

| Year | Reality | LSTM | | PSO-LSTM | | CAPSO-LSTM | |
|---|---|---|---|---|---|---|---|
| | | Prediction | Errors | Prediction | Errors | Prediction | Errors |
| 2017 | 8.350E+04 | 8.332E+04 | 0.22% | 8.298E+04 | 0.63% | 8.525E+04 | 2.09% |
| 2018 | 8.529E+04 | 8.507E+04 | 0.25% | 8.658E+04 | 1.52% | 8.898E+04 | 4.33% |
| 2019 | 8.870E+04 | 8.428E+04 | 4.99% | 8.513E+04 | 4.03% | 9.151E+04 | 3.17% |
| 2020 | 9.268E+04 | 7.995E+04 | 13.74% | 8.191E+04 | 11.62% | 9.067E+04 | 2.18% |
| 2021 | 9.313E+04 | 7.737E+04 | 16.92% | 8.236E+04 | 11.57% | 9.218E+04 | 1.02% |

## Evaluation criteria

The mean absolute error (MAE), root mean square error (RMSE), and mean absolute percentage error (MAPE) have been selected as the evaluative indices for the experimental outcomes of each algorithm. The equations for MAE, RMSE and MAPE are shown in Eq. (4).

$$
\begin{aligned}
\text{MAE} &= \frac{1}{n} \sum_{i=1}^{n} |\text{Pre}_i - \text{Rea}_i| \\
\text{RMSE} &= \sqrt{\frac{1}{n} \sum_{i=1}^{n} (\text{Pre}_i - \text{Rea}_i)^2} \\
\text{MAPE} &= \frac{1}{n} \sum_{i=1}^{n} \left| \frac{\text{Pre}_i - \text{Rea}_i}{\text{Rea}_i} \right|
\end{aligned}
\tag{4}
$$

where $n$ means the total number of sample, $\text{Pre}_i$ means the prediction value of $i$ and Rea means the reality value of $i$.

## RESULTS AND DISCUSSION

Prediction results are shown in Table 4. In the year 2017, the LSTM model exhibited the least prediction error at 0.22%, but this error margin escalated progressively over the years, culminating at 16.92% in 2021. This trend may indicate a degradation in the LSTM model's long-term prediction capability or its inability to adapt effectively to evolving trends in the data.

The PSO-LSTM model shows an initial error of 0.63% in 2017, increasing to 11.57% by 2021. Despite its fluctuating error rates across the observed years, the PSO-LSTM generally outperforms the standalone LSTM model, suggesting that Particle Swarm Optimization (PSO) contributes to a more refined parameter selection for the LSTM, thus improving its predictive accuracy.

Overall, the CAPSO-LSTM model exhibits consistently stable and superior predictive performance, with error rates decreasing from 2.09% in 2017 to 1.02% in 2021. This improvement signifies the success of the CAPSO in optimizing the hyperparameters of the LSTM, thereby enhancing the model's ability to capture the trends in urban green space area changes. Notably, in the year 2021, the CAPSO-LSTM model's prediction error is

**Table 5 Results of mean absolute error (MAE), root mean square error (RMSE) and mean absolute percentage error (MAPE).**

| Indicators | LSTM | PSO-LSTM | CAPSO-LSTM |
|---|---|---|---|
| MAE | 6,661.71 | 5,386.48 | 2,242.83 |
| RMSE | 9,272.96 | 7,024.29 | 2,431.38 |
| MAPE | 0.07 | 0.06 | 0.03 |

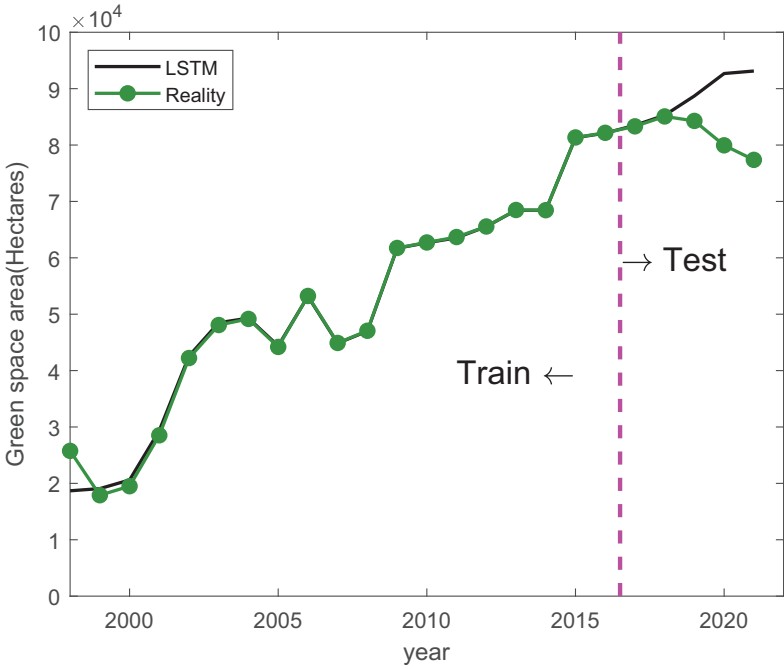

**Figure 3 Train and prediction results of LSTM.** The black line indicates the predicted values generated by an LSTM model, while the green line represents the actual observed values.

significantly lower than that of the other models, highlighting its potential and practicality in handling such data.

Results of mean absolute error (MAE), root mean square error (RMSE) and mean absolute percentage error (MAPE) are shown in Table 5. The performance of the LSTM, PSO-LSTM, and CAPSO-LSTM models was comprehensively evaluated. It was observed that the CAPSO-LSTM model consistently outperformed the other models, as evidenced by its significantly lower MAE (2,242.83), RMSE (2,431.38), and MAPE (0.03), indicating a superior predictive accuracy. The improvements in forecasting precision can be attributed to the optimization of hyperparameters using the CAPSO algorithm, which has been demonstrated to enhance the LSTM model's capability to predict urban green space areas more reliably. Conversely, the LSTM model, devoid of optimization techniques, was found to have the highest error rates, suggesting a lesser degree of reliability in its predictions. The PSO-LSTM model, which utilized Particle Swarm Optimization, showed a moderate performance improvement over the standard LSTM model, as reflected by its intermediate

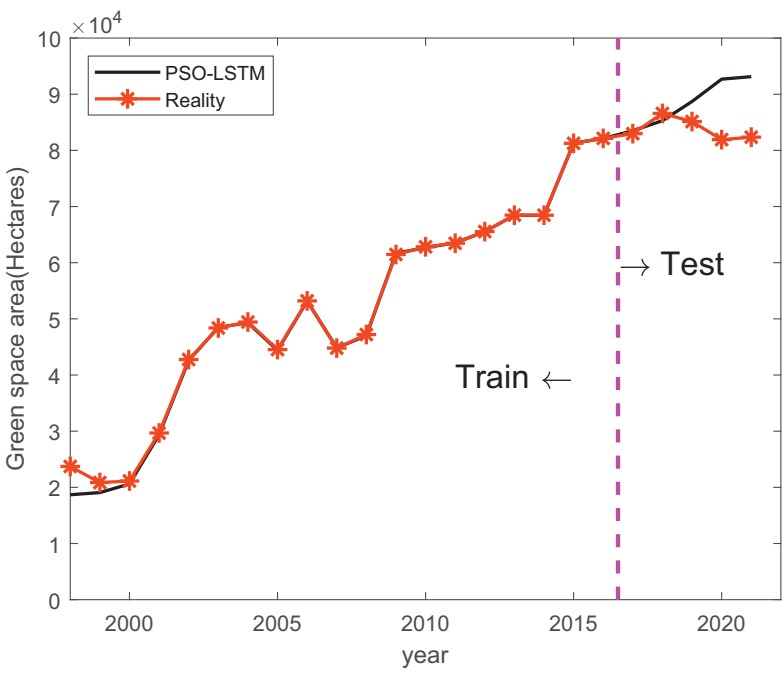

**Figure 4 Train and prediction result of PSO-LSTM.** The black line indicates the predicted values generated by an PSO-LSTM model, while the orange line represents the actual observed values.

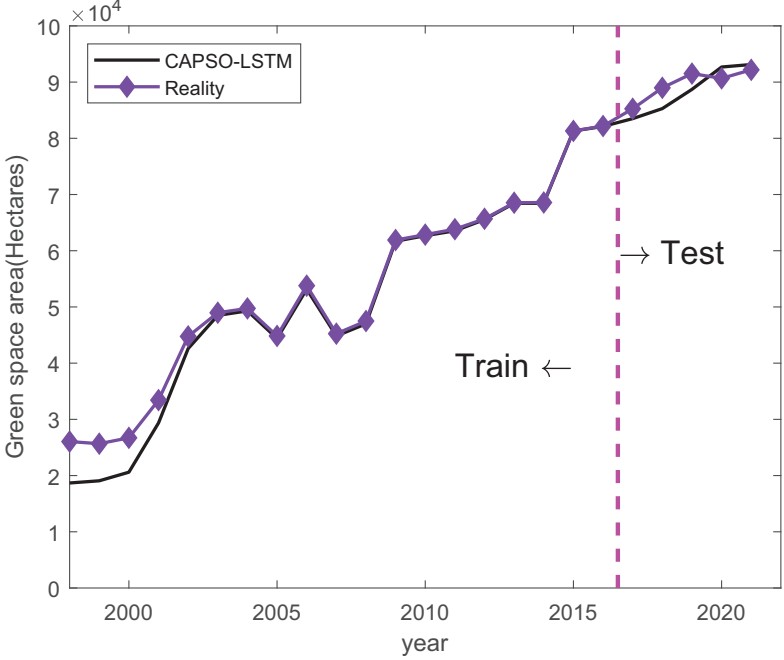

**Figure 5 Train and prediction result of CAPSO-LSTM.** The black line indicates the predicted values generated by an CAPSO-LSTM model, while the purple line represents the actual observed values.

values of MAE and RMSE, and a lower MAPE than that of the LSTM model. Train and prediction results of LSTM, PSO-LSTM, and CAPSO-LSTM are shown in Figs. 3–5.

Compared to traditional metaheuristic algorithms, CAPSO is specifically designed for the parameter training of recurrent neural networks, utilizing a cosine oscillation strategy to enhance the particle swarm algorithm's ability to escape local optima.

Moreover, the dataset for urban green space is collected on an annual basis, leading to a scarcity of training and testing data. When trained with such limited datasets, the CAPSO-LSTM model exhibits a distinct advantage over comparative algorithms. However, it is imperative to acknowledge that a deficiency in training data may precipitate a decline in model accuracy, accompanied by an escalation in both MAE and RMSE. This presents a formidable challenge to the CAPSO-LSTM framework, underscoring the importance of devising strategies for efficient training leveraging short-term data as a critical future research avenue.

In summary, the CAPSO-LSTM model significantly improves the accuracy of urban green space area predictions by optimizing the hyperparameters of the LSTM. This has important practical applications in urban planning and green space management, as accurate predictions can assist urban planners in making better-informed decisions to maintain or increase urban greenery.

## CONCLUSION

The imperative of accurately forecasting urban green space areas is acknowledged for its critical role in environmental conservation, urban planning, and ensuring societal well-being in urban areas. In this study, a new predictive model termed CAPSO-LSTM has been proposed, utilizing the CAPSO algorithm to fine-tune the hyperparameters of an LSTM network, thereby augmenting the model's performance.

In the conducted simulation experiments, the CAPSO-LSTM model's performance was found to be superior to that of both the PSO-LSTM and the traditional LSTM models. The specific experimental outcomes revealed that the CAPSO-LSTM model achieved an MAE of 2,242.83, an RMSE of 2,431.38, and a MAPE of 0.03.

However, it is recognized that the scarcity of training data, due to the annual collection cycle of urban green space-related data, has resulted in elevated MAE and RMSE among the algorithms tested. Based on these experimental outcomes, the development of predictive models capable of achieving high accuracy with limited training samples is identified as a critical area for future research.

In conclusion, the introduction of the CAPSO-LSTM model marks a significant step forward in the predictive modeling of urban green spaces. The employment of the CAPSO algorithm for hyperparameter optimization within the LSTM framework has proven effective, as evidenced by the simulation results. Future work should aim to address the identified limitations by exploring more computationally efficient algorithms that maintain the model's predictive accuracy while minimizing resource consumption. Additionally, further studies could investigate the model's adaptability to various urban contexts and the integration of additional predictive variables, such as climate patterns or urbanization rates, to enhance the model's robustness and applicability to real-world

scenarios. This future research will be critical for advancing the practical application of the CAPSO-LSTM model in sustainable urban development and green space management.

### Funding

This work was supported by the Natural Science Foundation of Hubei Province (2023AFB003) and the Education Department Scientific Research Program Project of Hubei Province of China (Q20222208). The funders had no role in study design, data collection and analysis, decision to publish, or preparation of the manuscript.

### Grant Disclosures

The following grant information was disclosed by the authors:
Natural Science Foundation of Hubei Province: 2023AFB003.
Education Department Scientific Research Program Project of Hubei Province of China: Q20222208.

### Competing Interests

The authors declare that they have no competing interests. Ke Yan is employed by Construction Third Engineering Bureau Installation Engineering Co., Ltd.

### Author Contributions

- Hao Tian conceived and designed the experiments, performed the experiments, prepared figures and/or tables, authored or reviewed drafts of the article, and approved the final draft.
- Hao Yuan conceived and designed the experiments, analyzed the data, performed the computation work, prepared figures and/or tables, and approved the final draft.
- Ke Yan performed the experiments, analyzed the data, authored or reviewed drafts of the article, and approved the final draft.
- Jia Guo conceived and designed the experiments, performed the experiments, analyzed the data, performed the computation work, authored or reviewed drafts of the article, and approved the final draft.

### Data Availability

The data is available at GitHub and Zenodo:
- https://github.com/GuoJia-Lab-AI/CAPSO-LSTM.
- Guo. (2024). GuoJia-Lab-AI/CAPSO-LSTM: Data (v1.0.0). Zenodo. https://doi.org/10.5281/zenodo.10512386.

### Supplemental Information

Supplemental information for this article can be found online at http://dx.doi.org/10.7717/peerj-cs.2048#supplemental-information.

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
