# Peer review of "A cosine adaptive particle swarm optimization based long-short term memory method for urban green area prediction"

_PeerJ Computer Science, doi:10.7717/peerj-cs.2048_

## Round 0.1 · original submission · Major Revisions

Dear authors,

Thank you for submitting your article. Feedback from the reviewers is now available. It is not recommended that your article be published in its current format. However, we strongly recommend that you address the issues raised by the reviewers, especially those related to readability, experimental design and validity, and resubmit your paper after making the necessary changes. In particular, Reviewer 1's reviews of metaheuristic optimization algorithms are important, and you are expected to address why the new algorithm would be beneficial to this area of research.

Reviewer 1 has suggested that you cite specific references. You are welcome to add it/them if you believe they are relevant. However, you are not required to include these citations, and if you do not include them, this will not influence my decision.

Best wishes,

Reviewer 1 ·

Basic reporting

The English used in the document is appropriate.
Some references of relevance to the study are missing.
The paper is well structured.
The paper is self-contained.
Some of the concepts presented in the paper need to be explained more clearly.

Experimental design

The paper is within the scope of the journal.
The topic addressed by the study is interesting.
It is necessary to include additional information to improve the study's rigor.
It is necessary to include additional information to improve the replicability of the study.

Validity of the findings

The novelty of the study is very limited
The data set used is provided; however, the proposed method has to be shown to achieve consistently good performance.
The conclusions are very superficial; the knowledge gap that has been closed should be discussed.

Additional comments

The study seeks to adapt a metaheuristic as an optimization algorithm in the training process of a neural network. The study is interesting but its degree of innovation seems to be low. Therefore, the following comments are made and a major revision is recommended.

Comments:
The study employs a metaheuristic as the optimization algorithm to train a neural network. This is a fairly common approach so it is imperative that the authors comment on the reason for using cosine adaptive particle swarm optimization instead of other widely known metaheuristics.

Because you address the topic of metaheuristics in your paper, it is imperative that you comment that dozens of new metaheuristics are proposed or even improved every year, as noted by Velasco et al. (doi: 10.1007/s11831-023-09975-0). Also, the thoroughness of the paper would be enhanced by commenting that these new metaheuristics are harshly criticized for their lack of innovation. More information can be found in Aranha et al. (doi: 10.1007/s11721-021-00202-9) and Sörensen (doi: 10.1111/itor.12001). What makes CAPSO an innovative algorithm? Please comment.

It appears that the authors used 17% of the data as the validation set, leaving the rest as training data. The 17% may be a low percentage to perform the validation process, being usual values closer to 25 or 35%. Can the authors comment on the reasons for using the mentioned percentage?

Due to their stochastic nature, metaheuristics always return different solutions when used. Additionally, as pointed out by Velasco, Guerrero, and Hospitaler (doi: 10.1016/j.swevo.2022.101172), the values of the hyperparameters of a metaheuristic affect its performance. Therefore, it is necessary to discuss the above, mentioning how the authors defined the hyperparameters of CAPSO, and provide some data to show that CAPSO can provide consistent performance on this problem.

The conclusions are very superficial. It is necessary to comment on the knowledge gap that was closed with this study. Likewise, some statements seem very propagandistic, such as the one presented between lines 227 and 228.

References
Aranha, C. et al. (2021) ‘Metaphor-based metaheuristics, a call for action: the elephant in the room’, Swarm Intelligence, (0123456789). doi: 10.1007/s11721-021-00202-9.
Sörensen, K. (2013) ‘Metaheuristics-the metaphor exposed’, International Transactions in Operational Research, 22, pp. 3–18. doi: 10.1111/itor.12001.
Velasco, L., Guerrero, H. and Hospitaler, A. (2022) ‘Can the global optimum of a combinatorial optimization problem be reliably estimated through extreme value theory?’, Swarm and Evolutionary Computation, 75. doi: 10.1016/j.swevo.2022.101172.
Velasco, L., Guerrero, H. and Hospitaler, A. (2023) ‘A Literature Review and Critical Analysis of Metaheuristics Recently Developed’, Archives of Computational Methods in Engineering. doi: 10.1007/s11831-023-09975-0.

Reviewer 2 ·

Basic reporting

1. The literature review is shallow.
2. The mathematical symbol is not written correctly in many places.

Experimental design

The work is original and new.

Validity of the findings

1. Novelty of the proposed work should be established by comparing the same with comparable work.
2. How do you decide the values of parameters in Table 3.
3. The values of MAE and RMSE are very high which indicates that the model's predictions are far off from actual values on average.
4. Proposed model should be compared to some conventional models.
5. Author should cite some recent work such as "Short-term traffic flow prediction based on optimized deep learning neural network: PSO-Bi-LSTM"

---

## Round 0.2 · accepted · Accept

Dear authors,

Thank you for the revision and for clearly addressing all the reviewers' comments. I confirm that the paper is improved and addresses the concerns of the reviewers. Your paper is now acceptable for publication in light of this revision.

Best wishes,

Reviewer 1 ·

Basic reporting

The English used in the document is appropriate.
The references are adequate.
The paper is well structured.
The paper is self-contained.
The concepts are clearly presented.

Experimental design

The paper is within the scope of the journal.
The topic addressed by the study is interesting.
No comment.
No comment.

Validity of the findings

The novelty of the study is very limited
The data set used is provided.
No comment.

Additional comments

The comments of this reviewer were answered in an appropriate manner. Therefore, the document is considered to be of adequate quality for publication.

Reviewer 2 ·

Basic reporting

Clear and unambiguous, professional

Experimental design

Original primary research within Aims and Scope of the journal.

Validity of the findings

All underlying data have been provided; they are robust, statistically sound, & controlled

Additional comments

The work is new and interesting.